# UCHL1 Regulates Radiation Lung Injury via Sphingosine Kinase-1

**DOI:** 10.3390/cells12192405

**Published:** 2023-10-05

**Authors:** Yulia Epshtein, Biji Mathew, Weiguo Chen, Jeffrey R. Jacobson

**Affiliations:** Department of Medicine, Division of Pulmonary, Critical Care, Sleep and Allergy, University of Illinois at Chicago, Chicago, IL 60612, USA; yuliaa@uic.edu (Y.E.); weiguo@uic.edu (W.C.)

**Keywords:** UCHL1, sphingosine kinase 1, radiation lung injury

## Abstract

GADD45a is a gene we previously reported as a mediator of responses to acute lung injury. GADD45a−/− mice express decreased Akt and increased Akt ubiquitination due to the reduced expression of UCHL1 (ubiquitin c-terminal hydrolase L1), a deubiquitinating enzyme, while GADD45a−/− mice have increased their susceptibility to radiation-induced lung injury (RILI). Separately, we have reported a role for sphingolipids in RILI, evidenced by the increased RILI susceptibility of SphK1−/− (sphingosine kinase 1) mice. A mechanistic link between UCHL1 and sphingolipid signaling in RILI is suggested by the known polyubiquitination of SphK1. Thus, we hypothesized that the regulation of SphK1 ubiquitination by UCHL1 mediates RILI. Initially, human lung endothelial cells (EC) subjected to radiation demonstrated a significant upregulation of UCHL1 and SphK1. The ubiquitination of EC SphK1 after radiation was confirmed via the immunoprecipitation of SphK1 and Western blotting for ubiquitin. Further, EC transfected with siRNA specifically for UCHL1 or pretreated with LDN-5744, as a UCHL1 inhibitor, prior to radiation were noted to have decreased ubiquitinated SphK1 in both conditions. Further, the inhibition of UCHL1 attenuated sphingolipid-mediated EC barrier enhancement was measured by transendothelial electrical resistance. Finally, LDN pretreatment significantly augmented murine RILI severity. Our data support the fact that the regulation of SphK1 expression after radiation is mediated by UCHL1. The modulation of UCHL1 affecting sphingolipid signaling may represent a novel RILI therapeutic strategy.

## 1. Introduction

Radiation-induced lung injury (RILI) is a potential sequela of radiation administered to the chest wall for which there are currently no targeted or effective therapies. The time course of RILI after radiation exposure is characterized by early pneumonitis (within 6–12 weeks), which is then followed by fibrosis at later time points (6–12 months). These events may be associated with significant functional impairment, which is evident at 3 mos and is progressive, with worsening lung function appreciable at 18 and 36 months [1]. A number of variables are known to contribute to RILI incidence after radiotherapy, including the nature of the underlying disease being treated, the particular radiation protocol utilized, and the concurrent administration of specific chemotherapeutic regimens [2,3]. Nonetheless, one study of over 800 patients with non-small cell cancer of the lung who were administered radiation reported pneumonitis associated with symptoms in 30% and fatal pneumonitis in 1.9% [4]. Notably, this incidence of fatality has since been corroborated by other studies [5,6]. The lack of effective RILI treatment is a significant barrier to care for these patients, and the identification of potential therapies could yield important clinical benefits to many patients with or at risk of RILI.

While mechanisms underlying RILI pathogenesis are poorly characterized, radiation is recognized to cause the stress-induced generation of reactive oxygen and nitrogen species, leading to cellular damage with apoptosis, inflammation, and dysregulated repair processes. Notably, pneumonitis associated with RILI is characterized by increased interstitial edema with the infiltration of inflammatory cells: features that are consistent with acute lung injury (ALI) are more generally known to be precipitated by increased endothelial cell (EC). We previously hypothesized that RILI pathobiology closely mimics that of other ALI syndromes and confirmed the critical role of sphingolipids in RILI, as evidenced by the differential expression of sphingolipids in vivo in RILI-challenged mice [7]. In addition, sphingosine kinase 1 (SphK1) knockout mice and mice with the reduced expression of specific sphingosine 1-phosphate (S1P) receptors were found to have increased RILI susceptibility. Further, we reported the protective effects of the S1PR1 receptor agonist, SEW-2871 (SEW) and (S)-FTY720-phosphonate (tyspinate), an S1P analog, in murine RILI [8].

Efforts to characterize the mechanisms underlying RILI mediated by sphingolipids have led to the identification of new molecular targets of interest in this context, including GADD45a (growth arrest and DNA damage repair gene 45 alpha) [9]. We reported that GADD45a−/− mice have an increased susceptibility to RILI, which is mediated by the decreased expression of UCHL1 (ubiquitin c-terminal hydrolase L1), a deubiquitinating enzyme, in these animals [9,10,11]. The known polyubiquitination of SphK1 suggests a possible mechanistic link between UCHL1 and sphingolipid signaling in RILI [12]. Thus, in the current work, we investigated the role of UCHL1 in sphingolipid-mediated RILI.

## 2. Materials and Methods

### 2.1. Antibodies and Reagents

Antibodies against UCHL1 (Cell Signaling, Danvers, MA, USA), SphK1 (Proteintech, Rosemont, IL, USA) phospho-SphK1 (ser-225) (ECM Biosciences, Versailles, KY, USA), Protein A-horse radish peroxidas (HRP) (Cell Signaling), and β-actin (Sigma, St. Louis, MO, USA), were purchased from the vendors indicated. Silencing RNA (siRNA) specific for UCHL1 and non-specific siRNA were both purchased from GE Dharmacon (Lafayette, CO, USA). The S1P analog, (S)-FTY720-phosphonate [(3S)-3-(amino)-3-(hydroxymethyl)-5-(4′-octylphenyl)-pentylphosphonic acid] (tysipinate) was synthesized as previously described and dissolved in ethanol [13]. Other reagents that were used include LDN-57444 (LDN, Calbiochem, San Diego, CA, USA) and MG-132 (Sigma).

### 2.2. Endothelial Cell Culture

Human pulmonary artery endothelial cells (EC) were cultured in an essential growth medium (EGM-2) containing 10% fetal bovine serum (Clonetics, Walkersville, MD, USA). EC was then allowed to grow to achieve confluent monolayers in an incubator at 37 °C, 5% CO_2_, and 95% humidity.

### 2.3. Endothelial Cell (EC) siRNA Transfection and Treatment with a UCHL1 Inhibitor

The transfection of EC with specific siRNA utilized serum-free conditions with DharmaFECT transfection reagent (Dharmacon, Lafayette, CO, USA) according to the manufacturer’s protocol. After 24 h, the medium was changed to EGM-2 (2% fetal bovine serum after). The silencing of the protein was then confirmed after 72 h, and all experiments utilizing silenced cells were conducted at 72 h. LDN, an inhibitor of UCHL1 inhibitor, was dissolved in 60% dimethylsulfoxide (DMSO). LDN (5 µM) was administered to cells for 1.5 h prior to experimentation with 60% DMSO used as a control.

### 2.4. UCHL1 Overexpression

To overexpress UCHL1 in human pulmonary EC, cells were transfected with human UCHL1 in a pcDNA3.1 vector (GenScript, Piscataway, NJ, USA) using the Lipofectamine 2000 transfection reagent (Invitrogen, Waltham, MA, USA) according to the manufacturer’s instructions. Cells were used after 36 h for experiments as described. The expression levels of UCHL1 in transfected cells were assessed using Western blotting.

### 2.5. Irradiation of EC

Human pulmonary artery EC was grown to confluence and then subjected to a single dose irradiation of 10 or 20 Gy using an X-Rad320 irradiator and a 320 kVp orthovoltage energy X-ray unit with a fully shielded cabinet, according to the manufacturer’s instructions (Precision X-Ray Irradiation, Madison, CT, USA). Cells were then used for specific experiments as described.

### 2.6. Immunoblotting and Immunoprecipitation

Protein extraction utilized an NP-40 lysis buffer (50 mM TrisHCl pH 7.4, 1% NP-40, 5 mM ethylenediaminetetraacetic acid, and 150 mM NaCl) with the addition of sodium fluoride 40 mM, phenylmethylsulfonyl fluoride 0.2 mM, sodium orthovanadate 0.1 M N’ ethyl malamide 10 mM, and a mixture of protease and phosphatase inhibitors (Calbiochem, San Diego, CA, USA). Sonicated lung homogenates and cell lysates were then subjected to thawing and freezing on dry ice. A bicinchoninic acid protein assay kit (Pierce, Rockford, IL, USA) was used to measure the concentration of proteins. Standard protocols were utilized for Western blotting, and ImageJ, http://imagej.nih.gov/ij/ (accessed on 9 May 2023), was utilized to quantify the density of individual bands. Immunoprecipitation utilized the RIPA buffer (Tris-HCL 50 mM, pH 7.4, NaCl 150 mM, 1% NP-40, 0.5% Sodium deoxycholate, 0.1% SDS, 1% NP-40, and a mixture of protease and phosphatase inhibitors) for protein extraction. Protein A/G Sepharose beads were then used for pre-clearing, with protein extracts incubated with primary antibodies overnight at 4 °C. Subsequently, immunocomplexes incubated with protein A/G Sepharose beads for 1 h at 4 °C and were then used for polyacrylamide gel electrophoresis with Western blotting performed subsequently.

### 2.7. Transendothelial Electrical Resistance (TER) Measurement

To measure TER, ECs were grown on gold-plated microelectrodes in polycarbonate wells containing evaporated gold microelectrodes with real-time measurements recorded using an electric cell-substrate impedance system (ECIS) (Applied Biophysics, Troy, NY, USA), as we have reported [14]. The EBM-2 medium containing 2% serum was used to grow cells to confluent monolayers, and measured resistance from each microelectrode was pooled at discrete time points and plotted versus time as the mean of the collective measurements. In these experiments, ECs were treated using tysipnate (25 ng/mL): an agonist that enhances barrier integrity.

### 2.8. Murine RILI Model

Male, 8- to 12-week-old male C57BL/6 mice (Jackson Laboratory, Bar Harbor, ME, USA) were utilized for the murine experiments. Briefly, animals were anesthetized with ketamine (150 mg/kg) and xylazine (15 mg/kg) via an intraperitoneal injection prior to being subjected to radiation administered as a single dose to the thorax (20 Gy), as we have previously described [8]. Mice were treated with LDN (5 mg/kg, IP) or a vehicle at the time of irradiation and then 3x/week for 6 wks. Bronchoalveolar lavage (BAL) fluid was used for the measurements of total protein and total cell counts consistent with our prior reports [10]. Approval by the University of Illinois at Chicago Animal Care and Use Committee was obtained for all animal experiments, and these experiments were conducted in full accordance with the Institute for Laboratory Animal Research Guide for the Care and Use of Laboratory Animals, Eighth Edition.

### 2.9. Statistics

Results are expressed as the mean ± SE. Student’s *t*-test was used for statistical analysis. Differences between experimental groups with a *p*-value less than 0.05 were considered statistically significant.

## 3. Results

### 3.1. Effect of Radiation on Lung EC and SphK1 Expression

In the initial experiments, we investigated the differential effects of radiation on UCHL1 and SphK1 protein levels in lung ECs. Human pulmonary artery ECs were subjected to irradiation (10 or 20 Gy) for 4 or 24 h. Lysates were then collected and used for Western blotting. These experiments confirmed a significant increase in both UCHL1 and SphK1 with both doses of radiation, as early as 4 h and persisting at 24 h (Figure 1).

### 3.2. Role of UCHL1 in Radiation-Induced SphK1 Ubiquitination and Expression Levels

As SphK1 is known to be polyubiquitinated and a precursor to proteasomal degradation affecting decreased protein levels, we examined the effects of radiation on SphK1 ubiquitination and the role of UCHL1 in this context. Human pulmonary artery ECs were subjected to irradiation (20 Gy, 6 h) and lysates were then used for IP for SphK1 followed by Western blotting for ubiquitin with either a standard anti-mouse horse radish peroxidase (HRP)-conjugated antibody or an HRP-conjugated protein A: a technique which results in reduced background signal via binding only to intact with IgG molecules of interest and not residual denatured heavy and light chain polypeptides of the IP antibody [15]. These experiments confirmed increased SphK1 ubiquitination after radiation with multiple bands evident, which is consistent with mono and poly-ubiquitination of SphK1 (Figure 2A). In subsequent experiments, ECs were either transfected with silencing RNA specific for UCHL1 (siUCHL1) or treated with LDN-5744 (LDN), a pharmacologic inhibitor of UCHL1 activity, prior to irradiation (20 Gy, 6 h) followed by IP of ubiquitin and Western blotting for SphK1. These experiments demonstrate an abrogation of radiation-induced SphK1 ubiquitination associated with both UCHL1 knockdown and inhibition (Figure 2B,C).

### 3.3. UCHL1 and SphK1 Interdependence

To further explore the link between UCHL1 and SphK1, we conducted complementary experiments to characterize SphK1 expression in response to differential UCHL1 expression levels and, in separate experiments, UCHL1 expression in response to differential SphK1 phosphorylation. Human pulmonary artery ECs were transfected with a siUCHL1 or a UCHL1 overexpression vector and lysates before being used for Western blotting for SphK1. Separately, ECs were treated with PF-543, a pharmacologic SphK1 inhibitor, and then subjected to irradiation (10 Gy, 24 h) prior to the Western blotting of lysates for UCHL1 (Figure 3). SphK1 expression levels closely correlated with both decreased and increased UCHL1 expression levels. Further, while radiation-induced significant SphK1 phosphorylation, the inhibition of SphK1 phosphorylation was associated with a marked reduction in radiation-induced UCHL1 expression levels.

### 3.4. Role of UCHL1 in EC Barrier Function after Radiation

To investigate the functional role of UCHL1 in EC barrier function after radiation, human pulmonary artery ECs were grown to confluence, overlying gold-plated microelectrodes to measure transendothelial electrical resistance (TER), and were then either transfected with siUCHL1 or treated with LDN prior to being subjected to irradiation. Cells were then treated with (S)-FTY720-phosphonate (tyspinate), a barrier-enhancing S1P analog, with a normalized resistance measure over time. These experiments confirmed a decrease in EC barrier enhancement induced by tyspinate after radiation pre-treatment. Further, both the knockdown of UCHL1 and inhibition of UCHL1 were associated with a significant decrease in EC barrier enhancement by tyspinate, both with and without radiation pre-treatment (Figure 4). These results support UCHL1 as an important regulator of sphingolipid-mediated EC barrier enhancement, both in general and after radiation. As SphK1 is known to enhance EC barrier function through the phosphorylation of sphingosine with a resultant increase in S1P expression [16], evidence of the attenuation of EC barrier enhancement induced by an S1P analog (tyspinate) after UCHL1 inhibition or UCHL1 knockdown highlights the functional intersect of these signaling pathways.

### 3.5. Role of UCHL1 In Vivo In Murine RILI

To study the role of UCHL1 in the elaboration of vascular permeability and inflammation associated with RILI in vivo, wildtype C57Bl/6 mice were treated with LDN (5 mg/kg, IP) or a vehicle at the time of and then for 6 weeks (3x/wk) after an RILI challenge with 20 Gy of single dose thoracic radiation. Whole lungs were harvested from selected vehicle-treated animals and used to assess changes in UCHL1 expression in response to radiation over time. UCHL1 mRNA levels were significantly increased 1 d after radiation, while both UCHL1 and SphK1 protein levels were significantly increased at 6 weeks (Figure 5).

Separately, BAL was collected at 6 weeks from animals in each experimental group and used to measure total protein levels and total cell counts as assessments of lung vascular permeability and inflammation. Radiation alone was associated with significant increases in both measures at 6 weeks (Figure 6). Treatment with LDN had no effect on control animals, while irradiated mice that were treated with LDN had an augmented injury response compared to irradiated controls, as assessed by both BAL protein and BAL cell counts; data are consistent with our in vitro findings.

## 4. Discussion

Our results confirm a significant upregulation of both UCHL1 and SphK1 in response to radiation, both in vitro in lung ECs and in vivo in whole lung homogenates from mice. Further, we observed that UCHL1 and SphK1 expression levels were interdependent on each other. UCHL1 attenuates the radiation-induced ubiquitination of SphK1 while inhibition of SphK1 attenuates the upregulation of UCHL1 after radiation. Finally, we found that UCHL1 mediates sphingolipid-induced EC barrier enhancement, while the inhibition of UCHL1 augments RILI in vivo. These data implicate UCHL1 as an important mediator of lung EC responses to radiation via the effects on sphingolipid signaling.

Sphingolipids are plasma membrane components that are recognized to mediate a wide range of cell-signaling events. S1P is a bioactive phospholipid produced largely by activated platelets as well as ECs themselves upon the phosphorylation of sphingosine by SphK1. Known to regulate EC signaling in a variety of contexts, S1P also promotes EC barrier function and attenuates lung vascular permeability in ALI models, including RILI [8,16,17]. Additionally, the activation of SphK1 also reduces lung vascular permeability in ALI through increased S1P signaling [18], while S1P analogs, including tyspinate, confer protection in murine RILI [8].

Collectively, our findings now are consistent with the idea that increased SphK1 expression levels after radiation are secondary to increased UCHL1 expression with the consequent deubiquitination of SphK1. Further, increased UCHL1 expression is itself dependent on SphK1, potentially through effects occurring in the nucleus, including histone acetylation that affects decreased DNA methylation (Figure 7). This possibility is suggested by known increased histone acetylation mediated by sphingosine kinases [19]. However, the point at which radiation initiates this positive feedback loop remains unclear. At the same time, S1P is derived from the phosphorylation of sphingosine via SphK1 and is released extracellularly [20,21]. The S1P ligation of specific S1P receptor subtypes (S1PR1-3) promotes EC barrier enhancement via cortical actin rearrangement, which is mediated by the small GTPase, Rac [14,22].

Functionally, UCHL1 removes or edits ubiquitin chains from ubiquitinated proteins affecting a variety of cellular processes including DNA repair, cell–cell communication, and apoptosis [23]. While the role of UCHL1 in health has not been well-characterized, the literature is rife with reports of its role in diseases, including neurologic disorders, such as Parkinson’s [24,25] and Alzheimer’s diseases [26], and in cancer biology, where it has been characterized as both an oncogenic factor [27,28,29] and as a tumor repressor [30,31,32,33]. However, we have previously reported that UCHL1 mediates lung vascular permeability in vitro and in a murine model of ventilator-induced lung injury [34]. UCHL1 in these studies attenuated lung vascular permeability via alterations in the adherents and tight junctional proteins, VE-cadherin, and claudin-5, respectively.

Although we found that UCHL1 is a mediator of lung vascular responses to radiation through effects on sphingolipids, alternative mechanisms of UCHL1 effects cannot be excluded. UCHL1 is a deubiquitinating enzyme known to have myriad targets, several of which might be relevant in this context. For example, UCHL1 deubiquitinates HIF-1α, which has been shown to mediate EC protection in response to radiation [35,36]. This literature supports increased EC resistance to radiation, which is associated with the upregulation of the vascular endothelial growth factor (VEGF) and basic fibroblast growth factor (bFGF) and regulated by increased HIF-1α expression and activation. Separately, UCHL1 also targets the TGF-β pathway via interactions with Smad2 and Smad3 [37]. UCHL-1 promotes TGF-β activation, while the depletion of UCHL-1 attenuates TGF-β signaling. This is notable as TGF-β levels have been linked to radiation-induced pulmonary fibrosis as a sequela of radiation pneumonitis [38,39]. In addition, EC UCHL1 has also been reported to attenuate the activity of TNF-α [40], an inflammatory cytokine that has itself been implicated as a key mediator of radiation pneumonitis [41]. Nonetheless, while it is likely that UCHL1 has multiple effects that are relevant to EC responses to radiation, it is clear that effects on sphingolipids are important in this context, a claim supported both by our data now and our previously published work [8].

The clinical burden of RILI is difficult to measure; however, a recent large retrospective study of hospitalization rates from radiotherapy complications provides some insights [42]. Upon a review of over 400,000,000 hospitalizations in the U.S. from 2005 to 2016, over 10,000 of these were for patients with RILI, which was associated with a mortality of 8%. Of course, these numbers do not account for the impact RILI has on those patients who survive hospitalization, including significant lung function impairment resulting in decreased functional capacity alongside the psychosocial impact on patients, reductions in quality of life, and critical interruptions in cancer therapy regiments. Taken together, RILI represents a clinical syndrome that affects a large number of patients, is associated with significant morbidity and mortality, and for which effective therapies are needed. While further studies are needed, our findings suggest that targeting the UCHL1-SphK1 signaling pathway may represent a novel therapeutic strategy for patients with or at risk of RILI.

## Figures and Tables

**Figure 1 cells-12-02405-f001:**
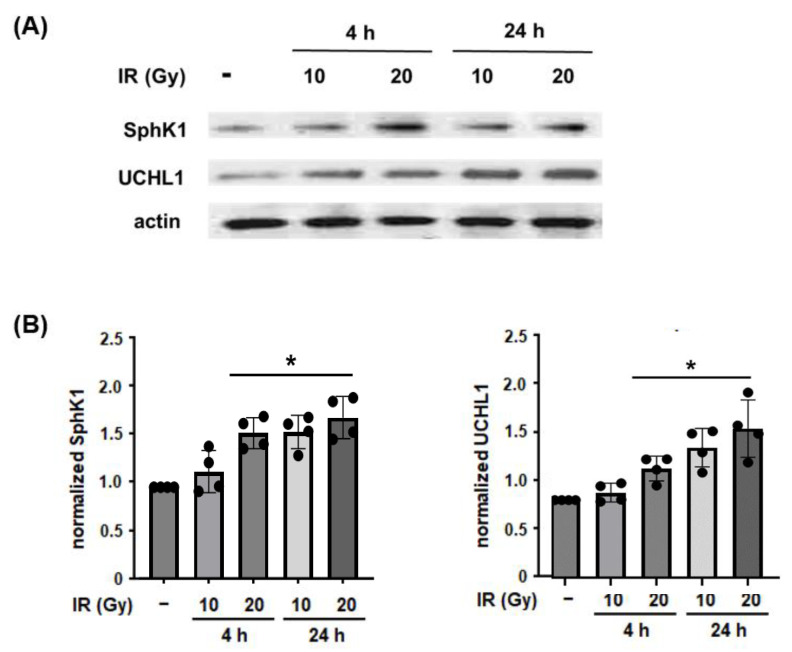
Radiation induces lung EC UCHL1 and SphK1 expression in vitro. Human pulmonary artery ECs were subjected to single-dose irradiation (10 or 20 Gy) for 4 or 24 h, and lysates were then subjected to Western blotting for SphK1 and UCHL1. (**A**) Representative blots shown with the same loading controls used for both SphK1 and UCHL1. (**B**) Densitometry from multiple experiments was performed for SphK1 and UCHL1, and both normalized to actin (n > 3 per condition, * *p* < 0.05 compared to control cells).

**Figure 2 cells-12-02405-f002:**
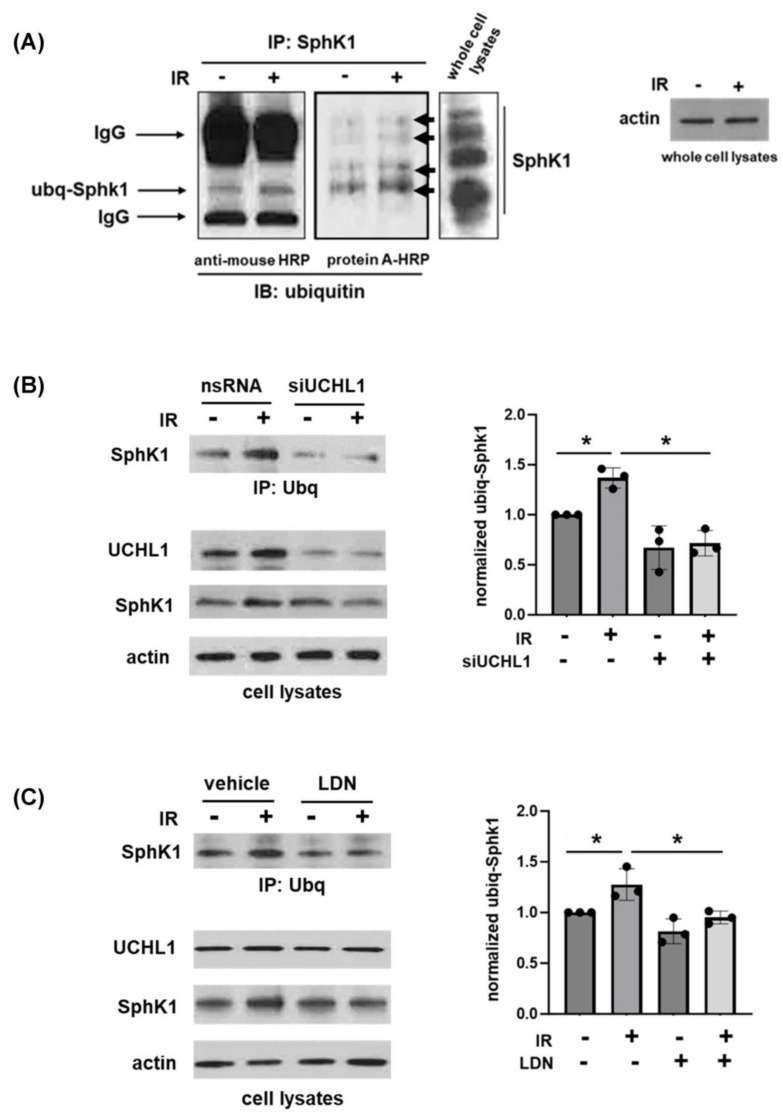
Radiation-induced SphK1 ubiquitination is regulated by UCHL1. (**A**) Human pulmonary artery ECs were subjected to radiation (IR, 20 Gy, 6 h) prior to immunoprecipitation with a SphK1 antibody followed by Western blotting for ubiquitin with either an anti-mouse secondary antibody (left panel) or HRP-conjugated protein A (middle panel). Whole-cell lysates from control cells were used for Western blotting for SphK1 (right panel) to confirm a predominant SphK1 band density at 50 kD. In separate experiments, human pulmonary artery ECs were (**B**) transfected with siRNA specific for UCHL1 (siUCHL1, 100 nM, 3 d) or non-specific siRNA (nsRNA) or (**C**) treated with LDN (5 µM, 1.5 h) prior to irradiation (20 Gy, 6 h) followed by the immunoprecipitation of SphK1 and Western blotting for ubiquitin. Representative blots shown. Densitometry quantified and expressed as ubq-SphK1 was normalized to total SphK1 (n = 3 per condition, * *p* < 0.05).

**Figure 3 cells-12-02405-f003:**
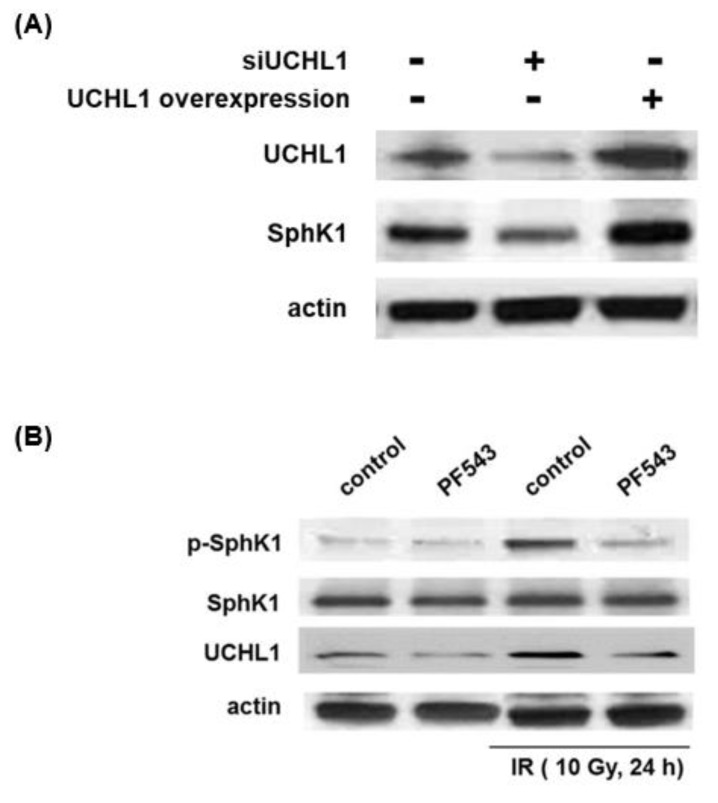
Expression levels of UCHL1 and SphK1 are interdependent in lung ECs. (**A**) Human pulmonary artery ECs were transfected with UCHL1 siRNA (100 nM, 3 d) or a UCHL1 overexpression vector (3 d) prior to Western blotting for SphK1. (**B**) In separate experiments, human pulmonary artery ECs were treated with a pharmacologic SphK1 inhibitor, PF-543 (10 µM), 1 h prior to irradiation, followed by Western blotting for UCHL1. Representative blots are shown.

**Figure 4 cells-12-02405-f004:**
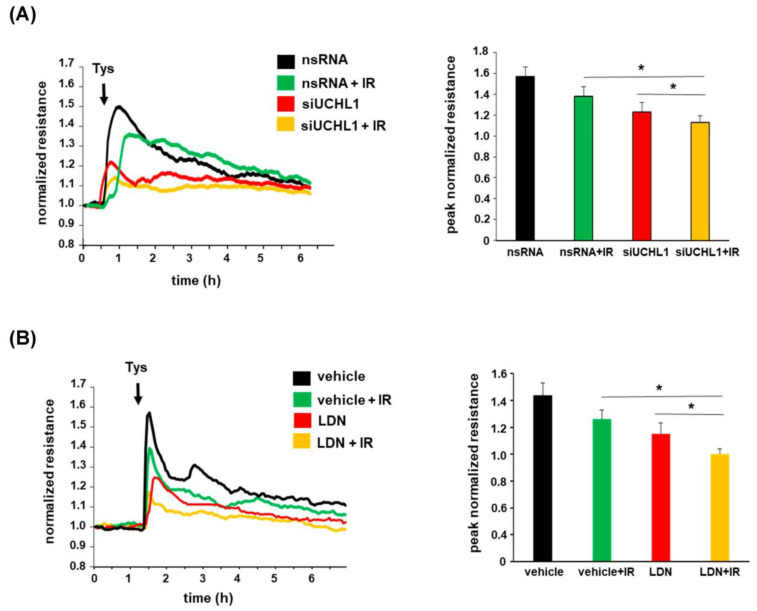
Lung EC barrier enhancement by tyspinate, an S1P analog, is attenuated by radiation and UCHL1 inhibition. (**A**) Human pulmonary artery ECs were transfected with siRNA specific for UCHL1 (siUCHL1, 100 nm, 3 d) or non-specific RNA (nsRNA) and grown to confluence on gold-plated microelectrodes prior to irradiation (20 Gy, 4 h) followed by treatment with tyspinate (Tys, 1 μM) and the measurement of transendothelial electrical resistance (TER). (**B**) In similar experiments, cells were subjected to irradiation (20 Gy, 4 h) prior to treatment with LDN (5 µM, 1.5 h) or a vehicle followed by tysipinate (Tys, 1 μM) (* *p* < 0.05, n = 3/experimental condition).

**Figure 5 cells-12-02405-f005:**
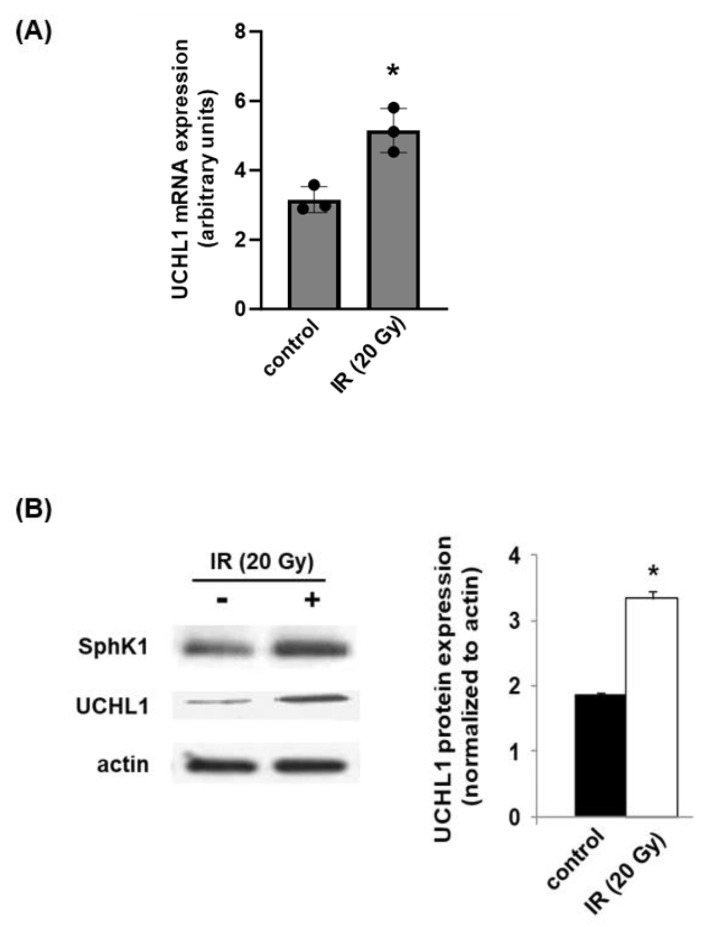
Radiation induces lung UCHL1 expression in vivo. (**A**) Male wild-type (WT) mice were subjected to RILI (20 Gy), and lungs were then harvested and used for RT-PCR to assess mRNA levels of UCHL1 at 6 wks (n = 3/group, * *p* < 0.05 compared to controls). (**B**) Lungs from RILI-challenged (20 Gy) WT mice at 6 wks were subjected to Western blotting for UCHL1 (n = 3 animals/group, * *p* < 0.05 compared to controls).

**Figure 6 cells-12-02405-f006:**
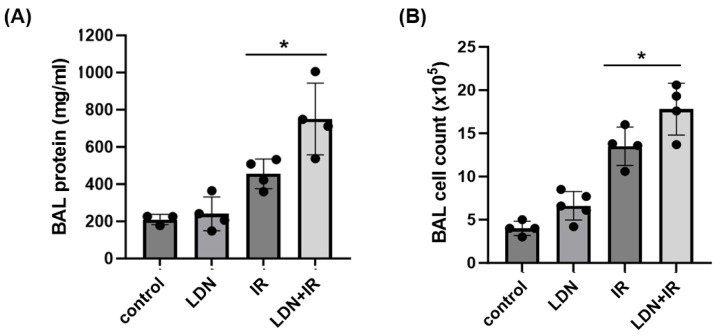
UCHL1 inhibition exacerbates RILI. WT mice were treated with LDN (5 mg/kg, IP) or a vehicle at the time of irradiation (IR, 20 Gy) and then 3x/wk post-radiation for 6 wks. BAL fluid was then collected and assessed for total protein (**A**) and total cell counts (**B**) (n ≥ 3/group, * *p* < 0.05 compared to respective controls, * *p* < 0.05).

**Figure 7 cells-12-02405-f007:**
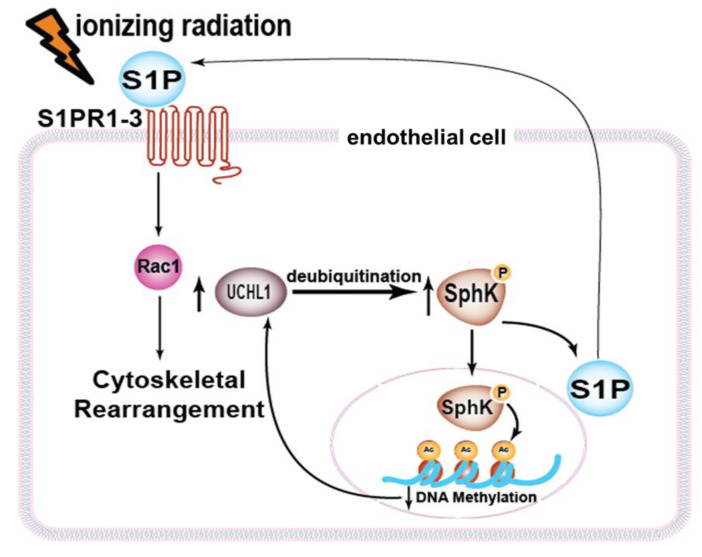
Proposed role of UCHL1 in sphingolipid-mediated responses to radiation. Our data are consistent with UCHL1-mediated EC barrier protection in response to radiation via decreased SphK1 ubiquitination which, in turn, promotes the increased activation of sphingosine 1-phosphate (S1P) as well as further increases in UCHL1 expression levels, potentially through effects on histone acetylation and DNA methylation. S1P is known to signal through S1P receptors on the surface of EC (S1PR1-3) to promote the activation of the small GTPase Rac1, actin cytoskeletal rearrangement, and EC barrier enhancement.

## Data Availability

The data presented in this study are openly available in FigShare at 10.6084/m9.figshare.23664030.

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
