# Peer review of "UCHL1 Regulates Radiation Lung Injury via Sphingosine Kinase-1"

_cells, 2023, doi:10.3390/cells12192405_

Round 1

Reviewer 1 Report

In this study, the authors tested the hypothesis that regulation of SphK1 ubiquitination by UCHL1 mediates radiation-induced lung injury (RILI). Human lung endothelial cells (EC) were subjected to radiation showed upregulation of UCHL1 and SphK1. The authors claimed that EC SphK1 ubiquitination was detected after radiation treatment (IR). Inhibition of UCHL1, using an inhibitor (LDN) or siRNA, attenuated sphingolipid-mediated EC barrier enhancement. Moreover, LDN pretreatment increased murine RILI severity. The authors proposed that regulation of SphK1 expression after radiation is mediated by UCHL1 and that modulation of UCHL1-mdiated sphingolipid signaling may represent a novel RILI therapeutic strategy.

        This study proposed a potential role of UCHL-mediated SphK1 regulation in EC response to radiation treatment; however, several issues need to be addressed to substantiate the authors conclusions. The specific comments are listed below.

Major comments:

1.     The increase of UCHL1 was not convincing, although the quantitative data presented increasing expression of UCHL1 (Fig. 1). It should be noted that the immunoblots of SPHK1 and UCHL1 shared the same loading controls (actin). The authors should present the image data that were consistent with the quantitative graphs.

2.     To support that UCHL1 and SPHK1 were increased at the proteins’ levels, the authors should also examine the mRNA expression levels of these two genes.

3.     The immunoblot of SPHK1 ubiquitination (Ub) upon IR was not convincing. The signal increase was exactly at the same position of SPHK1 protein. To the most, the data showed the mono (or multiple) ubiquitination, which may have function other than targeting for degradation, of SPHK1. To support that IR induced poly-Ub of SPHK1, the authors should have observed a significant mobility shift of SPHK1 tagged with Ub, as previously shown by  Loveridge C. et al. (JBC 285: 38841–38852, 2010).

4.     SPHK1 inhibitor PF543 pretreatment suppressed IR-induced increase of UCHL1. This figure (and other data) implicated that there could be a reciprocal regulation between SPHK1 and UCHL1. This data highlighted the contradictory descriptions the authors had stated between lines 254-257. The authors need to rephrase their Discussion.

5.     In Section3.4, the results presented in Figure 4 were weak in supporting the role of SPHK1 and UCHL1 in changes of barrier function induced by IR in ECs. The authors should have designed the experiment showing IR induced TER changes, which was further enhanced or suppressed by manipulations of SPHK1 and UCHL1 (levels or activities). TER changes induced by tysipinate, although very evident, was not relevant to the issue to be addressed here.

Minor comments:
6. Many typos and mistakes need to be corrected in the manuscript. Examples are listed below:

a.     In line 65, “phsoho-SphK1” should be “phospho-SphK1”.

b.     In line 152, should “ribosomal degradation” be “proteasomal degradation”?

c.     In line 195-196, “cells were the treated with…” should be “cells were then treated with…..”

d.     All abbreviations should be spelled out the first time appeared in the manuscript.

No additional comments

Author Response

Response to Reviewer 1

We are grateful  for the thoughtful and constructive comments of Reviewer 1.  In response to the comments of all of the reviewers, we have conducted additional experiments and we have carefully revised the text to provide clarity, to correct a number of typos, and to further address potential limitations of our study.  Our specific responses to each of the Reviewer 1 comments are detailed below.

Major comment 1: The increase of UCHL1 was not convincing, although the quantitative data presented increasing expression of UCHL1 (Fig. 1). It should be noted that the immunoblots of SPHK1 and UCHL1 shared the same loading controls (actin). The authors should present the image data that were consistent with the quantitative graphs.

Response to Major Comment 1:  We agree entirely with this comment and regret the choice of UCHL1 blots chosen for Fig 1A in the original submission of our manuscript.  We have now replaced it with one of the other original blots that is more representative of the cumulative data as presented in Fig 1B.

Major Comment 2:    To support that UCHL1 and SPHK1 were increased at the proteins’ levels, the authors should also examine the mRNA expression levels of these two genes.

Response to Major Comment 2:  Effects of radiation on mRNA levels of UCHL1 and SphK1 is an important line of investigation but is outside the focus of our studies which were aimed at differential ubiquitination of SphK1, accounting for differential SphK1 expression levels due to variable proteasomal degradation. For this reason, experiments designed to define mRNA expression levels of UCHL1 and SphK1 after radiation are likely to yield indeterminate results that neither support nor refute the significance of our protein data. The strength of our protein data is supported by consistent findings across multiple replicates, time points, and radiation dosing. Nonetheless, we appreciate this comment and have planned future studies to explore this area further.

Major Comment 3: The immunoblot of SPHK1 ubiquitination (Ub) upon IR was not convincing. The signal increase was exactly at the same position of SPHK1 protein. To the most, the data showed the mono (or multiple) ubiquitination, which may have function other than targeting for degradation, of SPHK1. To support that IR induced poly-Ub of SPHK1, the authors should have observed a significant mobility shift of SPHK1 tagged with Ub, as previously shown by  Loveridge C. et al. (JBC 285: 38841–38852, 2010).

Response to Major Comment 3:  This is a fair and fully justified comment. The single band highlighted in Fig 2A likely represents mono-ubiquitinated SphK1, given the lack of mobility shift as noted by the reviewer, while poly-ubiquitinated bands are obscured by the high background signal, a frequent problem with Western blots using IP samples due to detection of denatured heavy and light chains of the IP antibody. To address this, we conducted additional experiments using an HRP-conjugated protein A secondary antibody the detects only the intact IgG molecule of the primary antibody.  This technique is described by Lal et al and has now been cited in the text. The result, as now shown in the revised Fig 2A, confirms multiple increased bands after radiation consistent with mono- and poly-ubiquitinated SphK1.

Major Comment 4: SPHK1 inhibitor PF543 pretreatment suppressed IR-induced increase of UCHL1. This figure (and other data) implicated that there could be a reciprocal regulation between SPHK1 and UCHL1. This data highlighted the contradictory descriptions the authors had stated between lines 254-257. The authors need to rephrase their Discussion.

Response to Major Comment 4:  The reviewer is correct and we regret any confusion or language in the text that may have suggested the contrary. Our data do support reciprocal regulation of SphK1 and UCHL1 as noted. We have now revised the text to clearly highlight this fact in the discussion noting specifically that “UCHL1 and SphK1 expression levels are interdependent on each other”.  Further, the Figure 3 legend title highlights this idea (i.e. independence) as well and quantification for Fig 3B has been added.

Major Comment 5: In Section3.4, the results presented in Figure 4 were weak in supporting the role of SPHK1 and UCHL1 in changes of barrier function induced by IR in ECs. The authors should have designed the experiment showing IR induced TER changes, which was further enhanced or suppressed by manipulations of SPHK1 and UCHL1 (levels or activities). TER changes induced by tysipinate, although very evident, was not relevant to the issue to be addressed here.

Response to Major Comment 5:  This comment is entirely accurate in that the data presented in Fig 4 do not, by themselves, support a role of SphK1 in EC barrier function. We have now revised the text to clearly explain the justification for these experiments and to more accurately interpret the results.  Briefly, literature is now cited which supports the idea that EC barrier regulation by SphK1 is mediated by increased expression of S1P. Thus, evidence that EC barrier regulation in response to an S1P analog (tyspinate) is regulated by UCHL1 (as shown in Fig 4) suggests an important functional intersect between SphK1 and UCHL1 signaling pathways in this context.  We have made every effort to not overstate our results otherwise.

Minor comments:
6. Many typos and mistakes need to be corrected in the manuscript. Examples are listed below:

  1. In line 65, “phsoho-SphK1” should be “phospho-SphK1”.
  2. In line 152, should “ribosomal degradation” be “proteasomal degradation”?
  3. In line 195-196, “cells were the treated with…” should be “cells were then treated with…..”
  4. All abbreviations should be spelled out the first time appeared in the manuscript.

Response to Minor Comments:  We regret the several typos and mistakes that appeared in the original submission of our manuscript.  Each of the items above has now been corrected and all abbreviations are now spelled out the first time they appear. In addition, multiple authors have carefully read through the revised manuscript with additional corrections made as needed.

Reviewer 2 Report

In this study, the authors found SphK1 and UCHL1 were upregulated in both in vitro IR-treated EC cells and in vivo IR-treated mice. The regulation of SphK1 after IR is mediated by UCHL1 through sphingolipid signaling which may provide a new RILI therapeutic strategy. Although this research is important and interesting, some comments need to be addressed to improve the overall study:

1.    The author mentioned in line 142 the EC were subjected to IR for 4 or 24 hours. Did such conditions alter the cell phenotypes? Or change the cell proliferation rate or induce the cell death?

2.    The author indicate that SphK1 after IR is mediated by UCHL1 which is a deubiquitinated enzyme, however the ubiquitination of SphK1 still increased shown in figure 2A. Any explanations?

3.    In figure 2A, inputs should be shown. And did the author treat the cells with MG132 and detect the expression and ubiquitin of SphK1?

4.    In Figure 2B, the author also need to detect the expression level of SphK1 after MG132 treatment. And the input of SphK1 should be shown here.

5.    I am quite confused about the results shown in figure 3. In figure 3A, SphK1 is regulated by UCHL1 whenever UCHL1 is overexpression or knock-down. However UCHL1 was also decreased when treated with SphK1 inhibitor. Does author explain the mechanism which SphK1 regulate UCHL1?

6.    The English of this manuscript must be improved before resubmission. 

In this study, the authors found SphK1 and UCHL1 were upregulated in both in vitro IR-treated EC cells and in vivo IR-treated mice. The regulation of SphK1 after IR is mediated by UCHL1 through sphingolipid signaling which may provide a new RILI therapeutic strategy. Although this research is important and interesting, some comments need to be addressed to improve the overall study:

1.    The author mentioned in line 142 the EC were subjected to IR for 4 or 24 hours. Did such conditions alter the cell phenotypes? Or change the cell proliferation rate or induce the cell death?

2.    The author indicate that SphK1 after IR is mediated by UCHL1 which is a deubiquitinated enzyme, however the ubiquitination of SphK1 still increased shown in figure 2A. Any explanations?

3.    In figure 2A, inputs should be shown. And did the author treat the cells with MG132 and detect the expression and ubiquitin of SphK1?

4.    In Figure 2B, the author also need to detect the expression level of SphK1 after MG132 treatment. And the input of SphK1 should be shown here.

5.    I am quite confused about the results shown in figure 3. In figure 3A, SphK1 is regulated by UCHL1 whenever UCHL1 is overexpression or knock-down. However UCHL1 was also decreased when treated with SphK1 inhibitor. Does author explain the mechanism which SphK1 regulate UCHL1?

6.    The English of this manuscript must be improved before resubmission. 

Author Response

Response to Reviewer 2

We thank Reviewer 2 for providing the time and effort to carefully review our manuscript.  The concerns raised were constructive and fair. We have now conducted additional experiments in response and have addressed the bulk of these concerns in the revised text of the manuscript itself. Further, a direct response to each of the comments raised is provided below.

Major comment 1: The author mentioned in line 142 the EC were subjected to IR for 4 or 24 hours. Did such conditions alter the cell phenotypes? Or change the cell proliferation rate or induce the cell death?

Response to Major Comment 1:  This is an important point that we have now addressed in the revised text.  We did not observe changes in cell phenotype, proliferation, or cell death after the radiation dosing used for these studies.  Further, our findings are consistent with the literature that is also now cited in the text. In particular, Kabacik et al reported that endothelial cell monolayers remained intact and there was no increase in cell death at 30 h or 7 days after irradiation with doses as high as 10 Gy while <1% of endothelial cells were observed to be apoptotic in the central nervous system of mice at both 4 and 24 h after 50 Gy whole body irradiation (Pena et al).

Major Comment 2:    The author indicate that SphK1 after IR is mediated by UCHL1 which is a deubiquitinated enzyme, however the ubiquitination of SphK1 still increased shown in figure 2A. Any explanations?

Response to Major Comment 2:  We apologize for the confusion that was caused by the original presentation of our data in Fig 2A.  We have now performed additional experiments including Western blotting with an anti-protein A HRP antibody to reduce the abundant background signal.  These blots new more clearly demonstrate multiple ubq-SphK1 bands that are increased in response to radiation.

Major Comment 3: In figure 2A, inputs should be shown. And did the author treat the cells with MG132 and detect the expression and ubiquitin of SphK1?

Response to Major Comment 3:  We have now included the inputs for Fig 2A as requested.  In addition, we have performed new experiments using MG132 which confirm further increases in SphK1 after radiation associated with inhibition of proteasomal degradation (new Fig 2B).

 Major Comment 4: In Figure 2B, the author also need to detect the expression level of SphK1 after MG132 treatment. And the input of SphK1 should be shown here.

Response to Major Comment 4: As noted above, new data is now presented utilizing MG132 to inhibit proteasomal degradation of SphK1.  We have also revised this figure to include the input of SphK1 as requested (new Fig 2C and 2D).

Major Comment 5: I am quite confused about the results shown in figure 3. In figure 3A, SphK1 is regulated by UCHL1 whenever UCHL1 is overexpression or knock-down. However UCHL1 was also decreased when treated with SphK1 inhibitor. Does author explain the mechanism which SphK1 regulate UCHL1?

Response to Major Comment 5:  The reviewer has correctly interpreted our data as presented.  As stated in the text (Results 3.3 and Discussion), our data indicate that expression levels of UCHL1 and SphK1 are interdependent on each other. While our results do not identify a clear mechanism for this interdependent regulation, Figure 7 offers a potential mechanism that is further described in the third paragraph of the Discussion:

“Collectively, our findings now are consistent with the idea that increased SphK1 expression levels after radiation are secondary to increased UCHL1 expression with consequent deubiquitination of SphK1. Further, increased UCHL1 expression is itself dependent on SphK1, potentially through effects in the nucleus including histone acetylation affecting decreased DNA methylation (Figure 7). This possibility is suggested by known increased histone acetylation mediated by sphingosine kinases [21].”

Major Comment 6: The English of this manuscript must be improved before resubmission. 

Response to Major Comment 6:  We regret the several typos and mistakes of syntax that appeared in the original submission of our manuscript.  These items above have now been corrected. In addition, multiple authors have carefully read through the revised manuscript with additional corrections made as needed.

Reviewer 3 Report

In the present manuscript, Epshtein et al. demonstrate that radiation induced SphK1 expression is regulated by UCHL1, which could open a new therapeutic strategy against RILI. Although the work is novel and of potential interest in the field, I have a few major concerns which need to be addressed. My comments are shown below:

Major points:

1.       Figure 1A-B: The quantification shown in Figure 1B does not reflect in the blots in Figure 1A for both SphK1 and UCHL1. Please provide different representative blots.

2.       As SphK1 is known to generate S1P, did the authors ever check the effect of S1P in this process?

3.       Figure 2: Please provide the quantifications for Figure 2B and 2C.

4.       Figure 3: Please provide the quantifications for Figure 3B.

5.       Which S1P receptor could be involved in Figure 4 experiments? The authors could knock down S1PR1-3 in the cells and see the effects.

6.       Does the addition of S1P directly to the cells affect UCHL1 expression?

7.       The authors could check the level of S1P in the cell supernatant by ELISA or radioactivity assay.

8.       Did the authors perform any time course study before performing Figure 5 experiments? Why the protein level expression of UCHL1 is so late (6 weeks) as compared to mRNA expression (measured on day 1)? Did the authors check the UCHL1 expression on day1 or similar time?

9.       The in vivo studies are not appropriately performed. It could be possible that the difference in BAL proteins/cells count (Figure 6A-B) is a result of altered cell proliferation due to the course of treatment. The appropriate way to measure the in vivo vascular permeability is by Evan’s blue dye leakage assay.

Minor points:

1.       Abstract: Line 18: ‘UCHL1-silenced EC…..’ does not make sense. Please reframe it.

2.       Page 6, line 196: Typo: ‘the treated’ should be then treated.

Author Response

Response to Reviewer 3

We appreciate the constructive comments of Reviewer 3.  We have now made specific revisions to our manuscript in response to these comments which we believe greatly strengthen its merits.  Beyond these changes, however, several interesting and relevant points were raised that warrant consideration and are addressed in the revised Discussion as well. Below are our specific responses to each of the comments from this reviewer.

Major comment 1: Figure 1A-B: The quantification shown in Figure 1B does not reflect in the blots in Figure 1A for both SphK1 and UCHL1. Please provide different representative blots.

Response to Major Comment 1:  We fully agree with this comment and have revised Figure 1A to include blots that are more representative of the cumulative data as presented in Fig 1B.

Major Comment 2:    As SphK1 is known to generate S1P, did the authors ever check the effect of S1P in this process?

Response to Major Comment 2:  This is an important point. It is certainly the case that SphK1 is known to generate S1P.  It is also well established that S1P is a potent mediator of EC activation, signaling, and barrier regulation. While the current studies are focused on the SphK1-UCHL1 signaling axis and their interdependence, further exploration of S1P generation, as it relates to this axis, and its downstream effects is needed.  We have now added this idea to the Discussion in response to this comment and experiments are planned to extend our future investigations into this area.

Major Comment 3: Figure 2: Please provide the quantifications for Figure 2B and 2C.

Major Comment 4: Figure 3: Please provide the quantifications for Figure 3B.

Response to Major Comments 3 and 4:  These figures have now been revised in response to both of these comments with quantification now provided.

Major Comment 5: Which S1P receptor could be involved in Figure 4 experiments? The authors could knock down S1PR1-3 in the cells and see the effects.

Response to Major Comment 5:  Please see our response to Major Comment 2 above as it relates to this comment as well.  With respect to the specific S1P receptors involved, we have now added this consideration to the Discussion as well, cited relevant literature, and have planned future experiments to explore this very area.

Major Comment 6: Does the addition of S1P directly to the cells affect UCHL1 expression?

Response to Major Comment 6:  Please see our response to Major Comment 2 above as it relates to this comment as well.  While we did not perform these specific experiments, we have now proposed this very question in our Discussion and have planned future experiments to answer this question.

Major Comment 7: The authors could check the level of S1P in the cell supernatant by ELISA or radioactivity assay.

Response to Major Comment 7:  Please see our response to Major Comment 2 above as it relates to this comment as well.  We have now incorporated this question into the revised Discussion as well.  Taken together, all of these comments, related to S1P expression and signaling, represent an important line of investigation that lies outside the scope of our current studies but the revised manuscript now acknowledges these limitations and highlights the importance of these questions as areas of future investigation.  

Major Comment 8: Did the authors perform any time course study before performing Figure 5 experiments? Why the protein level expression of UCHL1 is so late (6 weeks) as compared to mRNA expression (measured on day 1)? Did the authors check the UCHL1 expression on day1 or similar time?

Response to Major Comment 8:  We agree that the original data as presented in Figure 5 raised additional questions related to the time course of UCHL1 mRNA and protein expression after radiation.  As it relates to the murine RILI model, our focus is specifically on UCHL1 expression levels and injury responses evident at 6 weeks, consistent with our prior publications utilizing this model. For this reason, we have now removed the data from day 1 from this figure.

Major Comment 9: The in vivo studies are not appropriately performed. It could be possible that the difference in BAL proteins/cells count (Figure 6A-B) is a result of altered cell proliferation due to the course of treatment. The appropriate way to measure the in vivo vascular permeability is by Evan’s blue dye leakage assay.

Response to Major Comment 9:  This is a fair point and we agree that, by itself, the BAL protein and cell count data may not reflect changes in vascular permeability alone. To address this concern, we have now added text which cites our previous work in which the RILI model was further characterized and confirmed significant increases in lung vascular permeability after radiation as measured by, not only BAL protein and cell counts, but also Evans Blue dye extravasation and significant changes evident by imaging (ViSen FMT) using an intravascular nontargeted blood pool probe (Mathew et al, AJRCMB 2011).

Minor Comments:

  1. Abstract: Line 18: ‘UCHL1-silenced EC…..’ does not make sense. Please reframe it.
  2. Page 6, line 196: Typo: ‘the treated’ should be then treated

Response to Minor Comments:  We regret the various typos and errors in syntax that appeared in the original manuscript submission. These have now been corrected and the entire manuscript carefully reviewed in an effort to identify any additional corrections needed.

Round 2

Reviewer 1 Report

The revision and the authors’ responses have answered most of the previous comments. However, few important additional changes need to be completed.

1.     The quality of the revised Fig. 2A is not satisfactory to show the significant ubiquitination of Sphk1. The authors may try transfection of an epitope-tagged ubiguitin to enhance the signal of Ub-Sphk1.

2.     Significant portions of the text in “Introduction” and “Materials and Methods” sections were copied from previous publications of the same group (e.g., PMID: 33438509, 34370281, 34571834). The authors should be noted about the copy right issues and make significant changes to the corresponding sections. An originality check should be done before another submission.

none

Author Response

We thank Reviewer 1 again for the additional construcitve comments and have tried to address them in the newly revised manuscript that is now submitted.  Briefly, we have repeated the experiments from the original Fig 2A and have include new blots that we believe better reflect the results described.  In addition, signficant changes were made to the text, specifically with respect to the Introduction and to the Materials and Methods section to mitigate potential copyright concerns.  We hope this now addresses the outstanding concerns of the reviewer.  Nonetheless, we welcome and appreciate any additional constructive comments as we know this will only help to improve the final manuscript.

Reviewer 2 Report

My comments are all addressed and now this manuscript is acceptable.

Author Response

We thank Reviewer 2 for the constructive comments in the initial review and are grateful that the revised manuscript was considered acceptable.

Reviewer 3 Report

The authors have tried to address all of my concerns. I am satisfied with the authors' responses and now after revision the manuscript is improved significantly. I do not have any further questions.

Round 3

Reviewer 1 Report

The revision has addressed previous concerns.